# Assessing the Clinical Relevance of Soluble PD-1 and PD-L1: A Multi-Cohort Study Across Diverse Tumor Types and Prognostic Implications

**DOI:** 10.3390/biomedicines13020500

**Published:** 2025-02-17

**Authors:** Nikolay E. Kushlinskii, Olga V. Kovaleva, Alexei N. Gratchev, Alexander A. Alferov, Yurii B. Kuzmin, Nikolai Y. Sokolov, Dmitry A. Tsekatunov, Irina B. Ryzhavskaya, Igor N. Kuznetsov, Dmitry N. Kushlinskii, Zaman Z. Mamedli, Ivan S. Stilidi

**Affiliations:** 1N.N. Blokhin National Medical Research Center of Oncology, Ministry of Health of the Russian Federation, 115478 Moscow, Russia; kne3108@gmail.com (N.E.K.); ovkovaleva@gmail.com (O.V.K.); alferov2a@ya.ru (A.A.A.); yriikuzmin@yandex.com (Y.B.K.); strivp@mail.ru (N.Y.S.); z.z.mamedli@gmail.com (Z.Z.M.); ronc@list.ru (I.S.S.); 2Ministry of Health of the Russian Federation, Russian University of Medicine, 127473 Moscow, Russia; npkredo@yandex.ru; 3Laboratory for Tumor Stromal Cells Biology, Institute for Carcinogenesis, N.N. Blokhin Cancer Research Center, Kashirskoye Sh. 24, 115478 Moscow, Russia; 4Regional State Budgetary Healthcare Institution of Khabarovsk Territory, Regional Clinical Oncology Center, 680042 Khabarovsk, Russia; dmtsekatunov@inbox.ru (D.A.T.); i9145417107@gmail.com (I.B.R.); drkushlinskiy@gmail.com (D.N.K.)

**Keywords:** sPD-1, sPD-L1, prognosis, cancer biomarkers, tumor progression

## Abstract

**Background/Objectives:** Immune checkpoint inhibitors targeting the PD-1/PD-L1 pathway have revolutionized cancer immunotherapy, however the clinical relevance of their soluble forms (sPD-1 and sPD-L1) remains less studied. Soluble PD-1 and PD-L1 have been implicated in tumor progression, prognosis, and treatment response across various malignancies. This study aims to provide a comprehensive analysis of sPD-1 and sPD-L1 levels in serum across diverse tumor types, including rare malignancies, and to evaluate their associations with clinicopathological characteristics and prognostic significance. **Methods:** In this study we analyzed sPD-1 and sPD-L1 levels in serum samples from 675 cancer patients representing a range of malignancies, including ovarian cancer, breast cancer, gastric cancer, colorectal cancer, renal cell carcinoma, and bone tumors. sPD-1 and sPD-L1 concentrations were measured using ELISA. Statistical analyses were performed to evaluate associations between soluble marker concentrations and clinicopathological factors, including tumor stage, size, histological subtype, and survival outcomes. **Results:** Elevated sPD-L1 levels were observed in several tumor types, including ovarian cancer, renal cell carcinoma, and gastric cancer, where they were associated with features of advanced disease, such as tumor size, stage, and metastases. In contrast, sPD-1 levels showed limited associations, with significant findings solely in gastric cancer and bone tumors, where levels correlated with histological subtype and differentiation. Prognostic analyses identified sPD-L1 as a marker of poor survival outcomes in ovarian cancer and bone tumors, while sPD-1 displayed no consistent prognostic significance. **Conclusions:** This study identifies the potential of sPD-L1 as a biomarker for tumor progression and prognosis across multiple malignancies. In contrast, sPD-1 showed limited clinical relevance, suggesting the importance of further investigation. These findings contribute to our understanding of soluble immune checkpoint proteins and their integration into personalized oncology strategies.

## 1. Introduction

One of the most significant advancements in cancer treatment in the past decade was the introduction of immune checkpoint inhibitors targeting CTLA-4 and PD-1/PD-L1 pathways. The first breakthrough came in 2011 with the approval of ipilimumab, a CTLA-4 blocking antibody [1]. This milestone was soon followed by the approval of monoclonal antibodies targeting PD-1 (pembrolizumab and nivolumab) and PD-L1 (atezolizumab and durvalumab), which expanded the range of immunotherapeutic options available to clinicians. Anti-PD-1 and PD-L1 antibodies demonstrated remarkable efficacy across a wide spectrum of malignancies, including melanoma, non-small cell lung cancer (NSCLC), renal cell carcinoma, head and neck cancers, and urothelial carcinoma [2]. Their success is attributed to their ability to restore T-cell function and enhance anti-tumor immune responses by blocking immune checkpoint pathways that tumors exploit to evade immune surveillance. Today, anti-PD-1/PD-L1 therapies are used either as monotherapy or in combination with chemotherapy, targeted therapy, or other immunomodulatory agents as first- or second-line treatments for nearly 50 types of malignancies [3,4].

PD-L1 is expressed on macrophages, activated T and B cells, dendritic cells (DCs), and certain epithelial cells, particularly under inflammatory conditions [5]. On tumor cells, PD-L1 expression is a key mechanism of immune evasion, enabling the suppression of anti-tumor immune responses. The presence of PD-L1 is often associated with an immune microenvironment enriched in CD8+ T cells, Th1 cytokines, chemokines, interferons, and specific gene expression profiles [6]. Interferon-gamma (IFN-γ), in particular, has been shown to induce PD-L1 expression in ovarian cancer cells, facilitating disease progression [7]. Notably, the inhibition of the IFN-γ receptor has been demonstrated to reduce PD-L1 expression, suggesting a therapeutic avenue for modulating this immune checkpoint pathway [8]. When PD-L1 binds to PD-1, it triggers PD-1-mediated signaling pathways within T cells, leading to the suppression of their proliferation, activation, and survival [9]. This signaling cascade effectively dampens the immune response, allowing tumor cells to evade immune-mediated destruction. The downstream effects include reduced cytokine production, metabolic reprogramming of T cells, and the promotion of an exhausted T-cell phenotype, which is characterized by diminished effector function [10].

This interplay between PD-1/PD-L1 and the tumor microenvironment highlights their pivotal role in immune escape and provides the rationale for therapeutic strategies targeting this axis. Immune checkpoint inhibitors, by blocking the PD-1/PD-L1 interaction, reactivate T cells, restoring their ability to develop an effective anti-tumor response. Understanding the regulatory mechanisms governing PD-L1 expression and PD-1 signaling is critical for optimizing these therapies and overcoming resistance in diverse cancer types. One such mechanism may involve the function of soluble forms of PD-1 and PD-L1, which are produced by cells expressing these immune checkpoint proteins.

The soluble form of PD-L1 (sPD-L1) is not membrane-bound or vesicle-associated but instead circulates freely in the bloodstream. Elevated levels of sPD-L1 have been detected in the serum of cancer patients [11], as well as in individuals with autoimmune diseases, viral infections [12], and during pregnancy [13]. This circulating form has been identified in more than 20 different pathologies, where it often plays a significant immunoregulatory role. In cancer, both sPD-1 and sPD-L1 have been detected in blood plasma, with elevated concentrations frequently correlating with disease progression and poorer clinical outcomes [14]. The soluble form of PD-1 (sPD-1) is thought to arise primarily through alternative splicing, while sPD-L1 is produced via proteolytic cleavage of membrane-bound PD-L1 by endogenous matrix metalloproteinases (MMPs) [15]. Experimental studies have shown that MMP inhibitors can suppress the secretion of sPD-L1 in the supernatant of PD-L1-transfected cell lines, underscoring the role of MMPs in its generation. Other enzymes, such as ADAM10 and ADAM17, have also been implicated in the cleavage of PD-L1, further expanding our understanding of the regulatory mechanisms behind its soluble form [16].

The role of sPD-L1 as a prognostic or predictive marker in cancers, such as lung cancer, has been the subject of intense debate. While its functional significance remains unclear, growing evidence suggests that sPD-L1 and sPD-1 may have critical roles in shaping immune responses and influencing therapeutic outcomes [17]. Further research is required to fully elucidate the mechanisms underlying these soluble checkpoint proteins and their potential as biomarkers in oncology and beyond.

In this study, we evaluated serum concentrations of the soluble forms of immune checkpoint proteins, sPD-1 and sPD-L1, across a diverse cohort of patients with malignant tumors. The analysis encompassed gastrointestinal malignancies, including gastric cancer (GC) and colorectal cancer (CRC); cancers of the female reproductive system, such as breast cancer (BC) and ovarian cancer (OC); and renal cell carcinoma (RCC), as well as malignant bone tumors. This investigation aimed to assess the clinical relevance of these soluble immune checkpoints and their potential associations with tumor characteristics and disease progression.

## 2. Materials and Methods

### 2.1. Patient Selection and Ethics Statement

This study included 267 patients with CRC, 101 patients with gastric cancer, 105 patients with RCC, 113 patients with malignant bone tumors, 67 patients with breast cancer, and 102 patients with ovarian cancer (Appendix A). The clinical diagnosis in all patients was confirmed through morphological examination of the tumor based on the actual WHO Classification of Tumors. The clinical utility of serum sPD-1 and sPD-L1 was studied in parallel to established laboratory parameters and the characteristics of the disease. Clinicopathological data were obtained from patient medical records.

Patients were eligible for this study if they were 18 years of age or older, had a histologically confirmed malignancy, and completed a written consent form for research using human derivatives, which allowed for secondary utilization of samples. A patient was excluded from this study if they had a diagnosis of two or more types of malignancy within the previous 5 years, active infections, a history of organ allograft, pregnancy or breastfeeding, received treatment before or during this study, or not enough samples were stored for analysis. Retrospective clinical and follow-up information was obtained from the medical records.

In addition, serum samples were collected from 175 healthy control subjects who were matched for age and sex and had no prior diagnoses of any malignancies and screened for acute inflammation and infection. These samples were used to assess the diagnostic utility of sPD-1 and sPD-L1. Before treatment, we collected blood serum samples using standard methods and stored them at −80 °C.

This study was approved by the Research Ethics Committee of N.N. Blokhin National Medical Research Center of Oncology (Approval no.: R2022-037, from 18 March 2022) and was performed in accordance with the Declaration of Helsinki. Written informed consent was obtained from all patients who provide samples.

### 2.2. ELISA

The concentrations of sPD-1 and sPD-L1 proteins were determined using ELISA in blood serum collected according to the standard method using EDTA before the start of specific treatment. Human PD-L1 ELISA Kit (BMS2327, Affymetrix, Santa Clara, CA, USA) and Human PD-1 ELISA kit (BMS2214, Affymetrix, Santa Clara, CA, USA) were used according to the manufacturer’s instructions. These kits use highly specific monoclonal antibodies for capture and detection, ensuring minimal cross-reactivity with related proteins. To validate the assay, standard curves were generated using recombinant protein standards provided in the kits, covering a dynamic range suitable for quantifying sPD-1 and sPD-L1 in serum samples. The sensitivity of the assays, as reported by the manufacturer, was 1.14 pg/mL for sPD-1 and 0.60 pg/mL for sPD-L1. All samples were measured in duplicate to ensure reproducibility, and intra- and inter-assay coefficients of variation (CV) were maintained within the acceptable limits recommended by the manufacturer. The measured proteins levels were expressed in picograms (pg) per milliliter (mL) of blood serum.

### 2.3. Statistical Analysis

The obtained data were processed using GraphPad Prism v. 10.4.1 software. For the selection of statistical analysis methods, the initial data were tested for normality using two criteria, the Kolmogorov–Smirnov test with the Lilliefors correction and the D’Agostino test. Since the data for sPD-1/sPD-L1 level did not follow a normal distribution, nonparametric tests were used. For comparisons of indicators and analysis of their relationships, nonparametric tests such as the Mann–Whitney U test, the Kruskal–Wallis test, and Spearman’s rank correlation coefficient were used. The diagnostic method’s informativeness, including sensitivity and specificity, was assessed by constructing ROC curves and calculating the area under the curve (AUC). Overall survival analysis was performed using the Kaplan–Meier method. The observation period spanned from the time of surgery to either the patient’s death or their last follow-up visit. Cutoff values for sPD-1 and sPD-L1 concentrations were defined as the median for each cancer type, so that survival analysis according to the soluble markers would not be affected by the potential difference in distributions of sPD-1 and sPD-L1 concentrations among cancer types. The statistical significance of differences between indicators was compared using the log-rank test. To evaluate the potential impact of various risk factors on survival, a multivariate analysis was additionally conducted using the nonparametric Cox proportional hazards model. Differences and correlations were considered statistically significant at *p* < 0.05.

## 3. Results

### 3.1. Study Population Overview

Serum samples were collected from a diverse cohort of cancer patients prior to the initiation of treatment. This study included 267 patients with colorectal cancer (CRC), 101 with gastric cancer, 105 with renal cell carcinoma (RCC), 113 with malignant bone tumors, 67 with breast cancer, and 102 with ovarian cancer. Detailed patient and group characteristics are presented in Appendix A. These tumor types were selected because they are known to involve immune regulation through the PD-1/PD-L1 pathway and represent a diverse set of cancers with varying immune microenvironments. Additionally, they are clinically relevant and prevalent, allowing for a comprehensive evaluation of sPD-1/sPD-L1 levels across different malignancies and their potential role in tumor progression and immune escape mechanisms. Importantly, none of the patients had conditions associated with known immunosuppression, such as a history of organ transplantation, human immunodeficiency virus (HIV) infection, or ongoing use of immunosuppressive medications.

### 3.2. Diagnostic Significance of sPD-1 and sPD-L1

The diagnostic significance of sPD-1 and sPD-L1 was assessed by comparing their levels across various patient groups and healthy controls. The median level of sPD-1 in the general healthy control group was 45.80 pg/mL (IQR: 33.58–57.31), which was significantly higher than in patients with gastric cancer (median: 11.82; IQR: 7.29–19.84). In contrast, it was significantly lower than in patients with renal cell cancer, where the median level was 58.78 pg/mL (IQR: 44.05–76.99). Among women in the healthy control group, the median sPD-1 level was 44.39 pg/mL (IQR: 31.59–56.02), which was markedly higher than in patients with breast cancer (median: 7.28; IQR: 5.84–9.04), while no significant difference was observed when compared to patients with ovarian cancer (median: 48.20; IQR: 35.76–62.78).

The median level of sPD-L1 in the general healthy control group was 9.01 pg/mL (IQR: 3.51–21.59), which was significantly lower than in groups of patients with all studied malignant tumors except colorectal cancer. The median level of sPD-L1 in the women healthy control group was 9.52 pg/mL (IQR: 3.64–29.22), which was significantly lower than in groups of patients with breast (median: 49.51; IQR: 39.10–81.86) and ovarian (median: 42.37; IQR: 17.21–83.09) cancer (Table 1, Figure 1).

Next, we performed a ROC (Receiver Operating Characteristic) analysis to evaluate the diagnostic performance of sPD-1 and sPD-L1 levels in various malignancies. The analysis demonstrated that sPD-1 and sPD-L1 show a wide range of discriminatory power, with AUC values from 0.540 to 0.989, reflecting their variable diagnostic utility in different tumor types. Notably, in the case of breast cancer, sPD-1 achieved an AUC of 0.989 (95% CI: 0.979–1.000, *p* < 0.0001), indicating excellent diagnostic accuracy, with sensitivity and specificity levels of 80.6% and 99.15%, respectively, at a concentration cutoff of 9.79 pg/mL. Moderate diagnostic accuracy was observed for sPD-L1 in breast and ovarian cancer with AUC values of 0.843 (95% CI: 0.786–0.900, *p* < 0.0001) and 0.735 (95% CI: 0.669–0.802, *p* < 0.0001), respectively. In gastric cancer, sPD-1 showed AUC of 0.945 (95% CI: 0.918–0.972, *p* < 0.0001); in contrast, sPD-L1 showed solely an AUC of 0.668 (95% CI: 0.601–0.737, *p* < 0.0001). In other tumors, both sPD-1 and sPD-L1 showed either no or low diagnostic accuracy. These findings indicate the variable performance of sPD-1 and sPD-L1 as diagnostic biomarkers, emphasizing their potential clinical utility in specific malignancies while underscoring the need for further investigation to refine their applicability (Figure 2).

### 3.3. Gynecological Cancers

#### 3.3.1. Breast Cancer

The analysis of sPD-1 and sPD-L1 levels in breast cancer patients revealed several key points. The levels of sPD-1 showed no significant differences across age groups, histological subtypes, tumor stages, sizes, nodal statuses, or grades. Similarly, sPD-L1 levels were generally consistent across most clinicopathological parameters (Table 2, Figure 3).

However, a notable exception was observed in HER2+ breast cancer, where sPD-L1 levels were significantly higher compared to luminal and triple-negative breast cancer (TNBC) subtypes (*p* = 0.035). The median sPD-L1 level for HER2+ tumors was 96.18 pg/mL (IQR: 65.60–313.10), almost double the levels observed in other subtypes, such as luminal (median: 48.24 pg/mL; IQR: 36.82–56.41) and TNBC (median: 51.53 pg/mL; IQR: 42.66–86.32). This finding points out the potential of sPD-L1 as a distinguishing biomarker for HER2+ breast cancer.

No other significant correlations were identified between the levels of sPD-1 or sPD-L1 and other clinicopathological characteristics, such as age, tumor stage, tumor size, nodal involvement, or grade. These results suggest that while sPD-L1 levels may have specific relevance in the context of HER2+ breast cancer, the overall utility of sPD-1 and sPD-L1 as broad biomarkers in breast cancer appears to be limited.

#### 3.3.2. Ovarian Cancer

The analysis of sPD-1 and sPD-L1 levels in ovarian cancer patients showed that serum sPD-1 levels were not significantly associated with any clinicopathological characteristics, including age, tumor histology, stage, size, nodal status, metastasis, or grade. This suggests a limited role for sPD-1 as a biomarker in ovarian cancer.

In contrast, sPD-L1 levels demonstrated significant associations with tumor size and disease stage. Patients with advanced-stage tumors (III–IV) had markedly higher median sPD-L1 levels (49.77 pg/mL; IQR: 29.10–100.60) compared to those with early-stage disease (I–II) (median: 17.88 pg/mL; IQR: 6.87–44.88; *p* < 0.0001). Similarly, larger tumors (T3–T4) were associated with significantly higher sPD-L1 levels (median: 49.77 pg/mL; IQR: 27.65–101.80) compared to smaller tumors (T1–T2) (median: 20.41 pg/mL; IQR: 7.03–46.87; *p* < 0.0001).

While sPD-L1 levels trended higher in patients with nodal involvement (N +) and metastases (M +), these differences did not reach statistical significance. Additionally, no significant associations were observed between sPD-L1 levels and age, histological subtype, or tumor grade (Table 3, Figure 3).

These findings suggest that while sPD-1 has no apparent clinical relevance in ovarian cancer, sPD-L1 may serve as a useful biomarker for tumor progression, particularly in relation to size and stage.

### 3.4. Immunogenic Tumors

#### Renal Cancer

Immunogenic tumors were represented in this study by renal cell carcinoma (RCC) of various histological types. The analysis revealed significant associations between the levels of sPD-1 and sPD-L1 and various clinical and pathological features in renal cell carcinoma (RCC). sPD-1 levels demonstrated a strong relationship with tumor histological type, with patients diagnosed with non-clear cell RCC (nccRCC) showing markedly higher median levels (87.01 pg/mL; IQR: 74.39–123.2) compared to those with clear cell RCC (ccRCC) (55.37 pg/mL; IQR: 40.62–72.64; *p* < 0.0001). Conversely, sPD-L1 levels did not show significant differences between these histological subtypes.

For sPD-L1, significant associations were observed with indicators of tumor progression. Higher sPD-L1 levels were found in patients with advanced disease stage (III–IV vs. I–II, *p* = 0.0004), larger tumor size (T3–T4 vs. T1–T2, *p* = 0.023), positive nodal status (N+ vs. N0, *p* = 0.0016), and the presence of metastases (M+ vs. M0, *p* = 0.009). Additionally, sPD-L1 levels were significantly elevated in tumors with lower differentiation (G3–G4 vs. G1–G2, *p* = 0.0451) (Table 4, Figure 3).

Demographic factors also influenced sPD-L1 levels. Older patients (over 60 years) exhibited significantly lower sPD-L1 concentrations compared to younger patients (*p* = 0.006), and males had higher sPD-L1 levels than females (*p* = 0.022). However, no such relationships were identified for sPD-1 in relation to age or gender (Table 4, Figure 3).

These findings highlight the potential utility of sPD-1 as a biomarker for distinguishing histological subtypes of RCC, while sPD-L1 appears to reflect tumor aggressiveness and progression.

### 3.5. Rare Tumors

#### Bone Tumors

Rare tumors included in this study are represented by several types of bone tumors. The analysis demonstrated that serum sPD-1 levels were significantly higher in patients with bone-forming tumors (osteosarcoma) compared to those with cartilage-forming tumors (chondrosarcoma, *p* < 0.0001). sPD-1 concentrations were also influenced by age and gender, with higher levels observed in patients younger than 40 years (*p* = 0.0246) and in males (*p* = 0.0165). Tumor grade further impacted sPD-1 levels, with significantly higher concentrations in high-grade tumors (G3) compared to lower-grade tumors (G1–G2, *p* = 0.0097) (Table 5, Figure 3).

In contrast, serum sPD-L1 levels showed fewer significant differences. However, they were notably higher in chondrosarcoma compared to osteosarcoma (*p* = 0.044) and were also significantly associated with tumor grade, with lower levels observed in high-grade tumors (G3) compared to lower-grade tumors (G1–G2, *p* = 0.0074). No significant associations were identified between sPD-L1 levels and other factors such as tumor size, stage, or metastasis (Table 5, Figure 3).

These findings suggest that sPD-1 and sPD-L1 levels may reflect distinct biological and clinical characteristics in bone tumors, with sPD-1 more strongly associated with patient demographics and disease progression, while sPD-L1 appears to be primarily influenced by tumor differentiation.

### 3.6. Gastrointestinal Cancer

Gastrointestinal tumors included in this study comprised gastric cancer and colorectal cancer.

#### 3.6.1. Gastric Cancer

The analysis of gastric cancer samples demonstrated significant associations between serum concentrations of sPD-1 and sPD-L1 with various clinicopathological characteristics. For sPD-L1, a statistically significant increase in median concentrations was observed with advanced tumor size (T3–T4 vs. T1–T2, *p* = 0.0003), higher disease stage (III–IV vs. I–II, *p* = 0.0011), and the presence of regional metastases (N+ vs. N0, *p* = 0.014). These findings suggest that sPD-L1 levels may serve as a biomarker for tumor progression (Table 6, Figure 3).

In terms of tumor histology, sPD-L1 concentrations were significantly higher in patients with signet ring cell carcinoma compared to adenocarcinoma (*p* = 0.049). Additionally, sPD-L1 levels increased as tumor differentiation decreased (G3 vs. G1–G2, *p* = 0.026), further reinforcing its association with aggressive disease features.

For sPD-1, a significant association was observed with tumor size, where higher levels were found in patients with smaller tumors (T1–T2 vs. T3–T4, *p* = 0.028). However, no statistically significant differences in sPD-1 levels were detected in relation to other factors such as age, gender, stage, nodal status, metastasis, or tumor grade. Notably, a significant correlation was identified between sPD-1 levels and the histological type of the tumor, particularly highlighting differences in gastric cancer subtypes (Table 6, Figure 3).

Overall, these findings underscore the potential clinical utility of sPD-1 and sPD-L1 as biomarkers for assessing tumor characteristics and disease progression in gastric cancer. The significant differences in sPD-L1 levels, particularly in relation to tumor size, stage, nodal status, and differentiation, point to its role in reflecting the aggressiveness of the disease. Conversely, sPD-1 appears to show more subtle associations, warranting further investigation into its potential diagnostic and prognostic value.

#### 3.6.2. Colorectal Cancer (CRC)

The analysis of sPD-1 and sPD-L1 levels in colorectal cancer patients revealed limited significant differences among the examined characteristics. Age showed a marginal association with sPD-1 levels, with patients aged over 62 having slightly higher median concentrations (40.53 pg/mL; IQR: 30.76–55.25) compared to those aged 62 or younger (35.56 pg/mL; IQR: 28.64–49.95; *p* = 0.041).

Regarding tumor grade, sPD-1 levels were significantly higher in G1 tumors (45.90 pg/mL; IQR: 31.01–59.99) compared to G2–G3 tumors (36.30 pg/mL; IQR: 29.51–49.94; *p* = 0.016). Similarly, sPD-L1 levels differed by tumor grade, with higher concentrations observed in G2–G3 tumors (7.34 pg/mL; IQR: 4.79–10.19) compared to G1 tumors (5.08 pg/mL; IQR: 3.44–9.19; *p* = 0.0046) (Table 7, Figure 3).

No statistically significant differences in sPD-1 or sPD-L1 levels were identified in relation to gender, tumor size, stage, nodal status, metastasis, or tumor localization. These findings suggest that while sPD-1 and sPD-L1 levels may reflect certain tumor characteristics such as grade, their association with other clinical factors in colorectal cancer appears limited.

#### 3.6.3. Prognostic Significance of sPD-1 and sPD-L1

To compare overall survival (OS) in patients with high vs. low sPD-1 and sPD-L1 serum levels, we used the Kaplan–Meier method. We observed no adverse outcome in the high sPD-1 groups of patients with studied malignant tumors. For PD-L1, it was shown that its high serum content is an unfavorable prognostic factor in RCC and ovarian cancer (HR = 4.34; *p* = 0.0038 and HR = 2.36; *p* = 0.0168, respectively) (see Table 8 and Figure 4).

Figure 4 illustrates the overall survival (OS) of patients stratified by high and low serum levels of sPD-1 and sPD-L1. Kaplan–Meier survival curves show that patients with renal cell carcinoma (RCC) and ovarian cancer (OC) who have high sPD-L1 levels experience significantly poorer OS compared to those with low sPD-L1 levels, emphasizing its potential role as an unfavorable prognostic factor. In contrast, no substantial differences in survival outcomes are observed between high and low sPD-1 groups, suggesting limited prognostic value for this marker, except in the case of bone tumors, where patients with lower sPD-1 levels showed slightly improved prognosis. These findings support the hypothesis of the clinical relevance of sPD-L1 as a biomarker for predicting survival outcomes, while the prognostic role of sPD-1 remains less clearly defined.

The Table 8 presents the results of univariate and multivariate Cox regression analyses assessing the prognostic significance of serum sPD-1 and sPD-L1 levels in various types of malignancies, focusing on overall survival (OS). Hazard ratios (HR), 95% confidence intervals (CI), and *p*-values are provided for both analyses, with significant findings highlighted.

For ovarian cancer, high sPD-L1 levels were significantly associated with poorer OS in both univariate (HR: 2.366; 95% CI: 1.179–4.750; *p* = 0.017) and multivariate (HR: 1.009; 95% CI: 1.000–1.018; *p* = 0.038) analyses, indicating that sPD-L1 is an independent prognostic factor. Conversely, sPD-1 levels showed no significant association with OS in either analysis.

In renal cancer, high sPD-L1 levels were significantly linked to worse OS in the univariate analysis (HR: 4.343; 95% CI: 1.846–10.22; *p* = 0.004), but this association was not maintained in the multivariate model (*p* = 0.368). sPD-1 levels were not significantly associated with OS in renal cancer.

For bone tumors, high sPD-L1 levels were identified as an independent prognostic factor for OS in the multivariate analysis (HR: 1.053; 95% CI: 1.000–1.004; *p* = 0.010). However, neither sPD-1 nor sPD-L1 levels showed significance in the univariate analysis.

In gastric cancer and colorectal cancer (CRC), neither sPD-1 nor sPD-L1 levels demonstrated a statistically significant association with OS in either univariate or multivariate analyses.

The results suggest that sPD-L1 may serve as an independent prognostic biomarker for OS in ovarian cancer and bone tumors, highlighting its potential clinical relevance in these malignancies. Other tumor types, including renal, gastric, and colorectal cancers, did not show consistent associations with OS based on sPD-1 or sPD-L1 levels.

### 3.7. Correlation Analysis

Next, we performed a correlation analysis to investigate the relationship between serum levels of sPD-1 and sPD-L1 across various malignancies and in healthy controls. The analysis revealed significant tumor-specific variations in the interplay between these immune checkpoint molecules. In colorectal cancer (CRC), a positive correlation was observed (R = 0.181, 95% CI: 0.059–0.299, *p* = 0.0029), suggesting linked regulatory mechanisms in this tumor type. Conversely, significant negative correlations were identified in gastric cancer (R = −0.605, 95% CI: −0.695 to −0.496, *p* < 0.0001) and breast cancer (R = −0.307, 95% CI: −0.478 to −0.113, *p* = 0.0018), indicating potentially distinct pathways driving the release of these soluble forms (Figure 5). In healthy donors, no significant correlation was observed (R = −0.010, *p* = 0.895), reflecting the absence of interplay between sPD-1 and sPD-L1 under normal physiological conditions. These findings suggest that the regulatory mechanisms governing sPD-1 and sPD-L1 expression and release are highly tumor-specific, emphasizing the complexity of their roles in cancer biology and the potential for differential biomarker utility across malignancies.

## 4. Discussion

Immune checkpoint inhibitors have profoundly transformed cancer immunotherapy, offering novel therapeutic options for a range of malignancies, including non-small cell lung cancer, renal cell carcinoma (RCC), melanoma, and triple-negative breast cancer (TNBC). The key points of this therapeutic revolution are the programmed cell death receptor 1 (PD-1) and its ligand (PD-L1), which serve as targets for immunotherapy. In addition to the membrane-bound forms, soluble forms of PD-1 and PD-L1 have also been detected in biological fluids. sPD-1 and sPD-L1 can be generated through multiple mechanisms, including alternative splicing, proteolytic cleavage by matrix metalloproteinases (MMPs), and exosomal secretion [18]. Different mechanisms are implicated in the generation of sPD-1 and sPD-L1 in different tumors. For example, PD-L1∆Ex5, which lacks exon 5, is described in various tumor types, with higher prevalence in lung, colon, and thyroid cancers compared to breast and pancreatic cancers [19]. Proteolytic cleavage by MMPs, such as MMP-13 and ADAM family proteases, also contributes to sPD-L1 release, with tumor-specific patterns of MMP expression influencing sPD-L1 levels [18,20]. Additionally, some cancers, including gastric and non-small cell lung cancer (NSCLC), show increased exosomal PD-L1 secretion [20]. The variability in sPD-1 and sPD-L1 production across tumors suggests distinct immunosuppressive landscapes that may affect responses to immune checkpoint inhibitors. However, key gaps remain, as most studies focus on single cancer types without direct comparisons across multiple malignancies, and mechanisms in bone tumors remain largely unexplored.

This variability in sPD-1 and sPD-L1 production across tumors not only reflects distinct immunosuppressive landscapes, but also has important implications for immune checkpoint inhibitor (ICI) therapy. While sPD-1 may enhance ICI efficacy by sequestering PD-L1 and preserving T-cell function, sPD-L1 can diminish therapeutic effectiveness by both suppressing T-cell activity and acting as a decoy for anti-PD-L1 antibodies [21,22,23]. These opposing roles underscore the need for further investigation into how soluble checkpoint molecules influence immunotherapy responses across different tumor types.

Despite numerous investigations, the diagnostic and prognostic value of sPD-1 and sPD-L1 remains not completely understood due to inconsistencies in the methodologies, limited tumor types studied, and small cohort sizes [18].

In our study, we utilized a standardized methodology to comprehensively analyze the levels of sPD-1 and sPD-L1 across a broad spectrum of malignancies, including rare bone tumors. Unlike prior research, which often focused on single tumor types or limited patient cohorts, our approach allowed for a broader evaluation of these biomarkers. This study revealed distinct tumor-specific patterns of sPD-1 and sPD-L1 expression and their associations with clinicopathological characteristics.

In the first phase of this study, we analyzed sPD-1 and sPD-L1 levels in patients with gynecological tumors. The sPD-L1 levels were significantly higher in ovarian cancer patients compared to the control group. Furthermore, we demonstrated that in ovarian cancer, sPD-L1 serves as a marker of poor prognosis, aligning with the existing literature. Specifically, studies have shown that both sPD-1 and sPD-L1 are unfavorable prognostic factors in ovarian cancer [24,25]. Additionally, sPD-L1 has been shown to help identify high-risk patients with poor outcomes, even in cases with platinum sensitivity, supporting the need for additional therapeutic approaches [26]. Our findings also revealed that the levels of sPD-L1 increase during tumor progression and spread. Evidence from the literature indicates that high levels of both sPD-1 and sPD-L1 in the serum of ovarian cancer patients are associated with advanced disease stage and poor prognosis, specifically reduced overall survival, and progression-free survival, which partially correlates with our results [25]. It is also worth noting that the levels of sPD-1 and sPD-L1 are not dependent on the degree of tumor cell differentiation in ovarian cancer, as confirmed by various studies [25,27].

In breast cancer (BC), we observed divergent trends in sPD-1 and sPD-L1 levels compared to healthy controls. Specifically, sPD-1 levels were significantly reduced, while sPD-L1 levels increased. Notably, HER2-positive BC patients exhibited significantly higher sPD-L1 levels than other subtypes. While our findings partially align with studies reporting elevated sPD-L1 levels in advanced BC, discrepancies exist regarding its association with histological subtypes [28]. Furthermore, the literature on tissue-based PD-L1 expression in BC reveals conflicting data, with some studies associating it with proliferative activity and differentiation grade, while others report no significant correlation [29]. Research on tissue expression of PD-1/PD-L1 in BC also yields contradictory results. While some studies report no association between these proteins’ expression and clinicopathological characteristics [30], others demonstrate a direct correlation between PD-L1 expression and proliferative activity or tumor cell differentiation grade [31]. Our findings suggest that the levels of sPD-1 and sPD-L1 do not correlate with clinicopathological characteristics, supporting the potential use of these analyses for diagnostic rather than monitoring purposes.

Our analysis of gastrointestinal tumors further highlights the differential roles of these biomarkers. In gastric cancer, sPD-1 levels were significantly lower, whereas sPD-L1 levels were elevated compared to healthy controls. Elevated sPD-L1 levels correlated with advanced tumor stage, size, and metastasis, suggesting its potential as a marker of tumor progression. These findings are consistent with studies linking high sPD-L1 levels to poor survival outcomes in gastric cancer [32]. However, the prognostic utility of sPD-1 remains ambiguous, with limited evidence supporting its role as an independent marker.

In colorectal cancer (CRC), a notable decrease in sPD-L1 levels was observed compared to healthy controls. However, no significant correlations were found with clinical characteristics or disease prognosis. These findings contrast with some reports identifying elevated sPD-L1 levels. Recent research has demonstrated that sPD-L1 levels are significantly elevated in CRC patients compared to healthy controls, indicating prognostic significance. For example, Shao et al. [33] performed enzyme-linked immunosorbent assay (ELISA) analyses in a large cohort of 300 CRC patients and 300 healthy individuals, identifying elevated sPD-L1 levels in CRC and their association with lymph node metastasis. Similar findings were reported by other groups, with studies confirming increased sPD-L1 levels in metastatic colorectal cancer (mCRC) and highlighting potential correlations with tumor burden and systemic inflammation [34,35]. Limited data also suggest elevations in sPD-1, though its role appears less well-defined and its clinical significance is inconsistent [35,36]. Further research is clearly needed to identify the diagnostic and prognostic roles of sPD-L1 in CRC.

Analysis of sPD-1 and sPD-L1 levels in renal cell carcinoma (RCC), a tumor responsive to checkpoint inhibitors, revealed no significant differences in sPD-1 levels between healthy donors and RCC patients, while sPD-L1 levels were significantly elevated. sPD-1 levels were associated with tumor histology, though sPD-L1 levels did not vary significantly across RCC subtypes. High sPD-L1 levels correlated with advanced disease stages, larger tumor sizes, metastases, and higher tumor grades, supporting its potential as a disease monitoring marker and prognostic indicator of poor outcomes [37]. However, some studies report no prognostic significance for sPD-L1 [38]. Elevated sPD-L1 levels have also been observed in metastatic clear-cell RCC and bladder cancer, further supporting its association with aggressive disease [21,39].

In the final phase of our study, we analyzed sPD-1 and sPD-L1 levels in patients with malignant bone tumors. Existing research on the diagnostic, clinical, and prognostic significance of these soluble forms in bone neoplasms is limited, with most studies not finding statistically significant correlations between these markers and clinicopathological factors or disease prognosis [40,41]. Our findings indicate that serum sPD-1 concentrations differed significantly between patients with chondrogenic (CS) and osteogenic (OS) tumors, with higher levels observed in the latter group. Additionally, we identified significant associations between the levels of these proteins and tumor cell differentiation. Specifically, poorly differentiated tumors exhibited elevated sPD-1 levels compared to well-differentiated ones, whereas sPD-L1 levels showed an inverse relationship.

While these soluble biomarkers are commonly used in clinical settings, their utility varies depending on the cancer type, stage, and individual patient characteristics. Some biomarkers, like CA-125 for ovarian cancer [42], and CEA and CA 19-9 for colorectal [43] and gastric cancer, are more commonly used for monitoring treatment response and detecting recurrence. For renal cancer VEGF, which is crucial in tumor angiogenesis and is often elevated in renal cell carcinoma (RCC), and CA 9 serve as prognostic markers and are used to predict responses, for example, to anti-angiogenic therapies [44,45]. However, none of these biomarkers are definitive on their own, and they are often used in combination with imaging, clinical evaluation, and other molecular tests for a more comprehensive diagnosis and prognosis. The role of soluble biomarkers is further enhanced when considered alongside novel markers like sPD-1/sPD-L1, which are involved in immune regulation and could add significant value to the existing biomarker panels by providing insights into immune response and tumor immune evasion mechanisms.

Despite the potential of sPD-1 and sPD-L1 as biomarkers, several challenges must be addressed before their integration into clinical practice. The cost of testing remains a significant issue, as no cost-effective immunoassays are currently available on the market. Further, variability in assay reproducibility across laboratories is a challenge for standardization, requiring extensive validation of methodologies. Incorporating these biomarkers into routine clinical workflows will require evidence-based guidelines and prospective studies to establish their reliability in diagnostic and prognostic decision-making.

Overall, our study underscores the variable clinical relevance of sPD-1 and sPD-L1 across tumor types. While sPD-L1 demonstrated consistent associations with markers of tumor aggressiveness and prognosis, sPD-1 exhibited limited and context-dependent relevance. These findings highlight the need for further research to refine the utility of these biomarkers in cancer management.

## 5. Study Limitations

This study has several limitations. First, the use of serum sPD-1/sPD-L1 levels as a proxy for tumor microenvironment dynamics was not directly evaluated. While serum biomarkers provide valuable systemic insights, they may not fully reflect local immune interactions within the tumor. Future studies should include tissue-based analyses to better understand the relationship between circulating and tumor-associated sPD-1/sPD-L1.

Second, the heterogeneity of the patient cohorts could impact the findings. This study included patients with different cancer types, disease stages, and blood sampling times, all of which may influence sPD-1/sPD-L1 levels.

Also, there were also no reference levels for sPD-L1, and the results between assay kits seemed to be quite different. There are also no pre-established cutoff levels that predict response or prognosis. To overcome this problem, some researchers are investigating reproducible, standardizable methods that can be used instead of ELISA.

## 6. Conclusions

The consistent application of a standardized pre-analytical method and ELISA for measuring sPD-1 and sPD-L1 levels across various tumor types in relatively large cohorts has provided valuable insights into the clinical utility of these soluble markers. Our findings demonstrate that sPD-1 exhibits diagnostic potential in breast and gastric cancers, where its levels are distinctively altered in patients compared to healthy controls. In contrast, sPD-L1 emerges as a robust marker for disease monitoring and prognosis, particularly in immunogenic tumors such as renal cell carcinoma, supported by our data and corroborated by the existing literature and unpublished findings in melanoma. Additionally, an unexpected and significant result was the utility of sPD-L1 in ovarian cancer, where it showed strong associations with tumor progression and poor prognosis, further highlighting its potential as a marker for risk stratification and therapeutic decision-making. Collectively, these findings emphasize the diverse roles of sPD-1 and sPD-L1 in oncology and underscore the importance of integrating these biomarkers into personalized diagnostic and monitoring strategies.

## Figures and Tables

**Figure 1 biomedicines-13-00500-f001:**
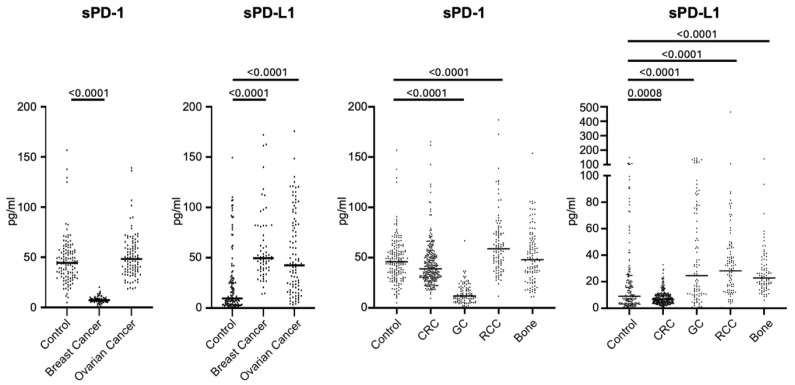
Comparison of serum levels of sPD-1 and sPD-L1 between patients with malignant tumors and healthy controls. Serum levels of sPD-1 and sPD-L1 were measured using ELISA and compared across different malignancies and healthy controls. sPD-1 levels were significantly lower in gastric and breast cancer patients compared to healthy controls but higher in renal cancer patients. In contrast, sPD-L1 levels were significantly elevated in all tumor groups except colorectal cancer. Notable differences were observed in female-specific cancers, with breast and ovarian cancer patients showing significantly higher sPD-L1 levels than healthy women. Lines represent median values.

**Figure 2 biomedicines-13-00500-f002:**
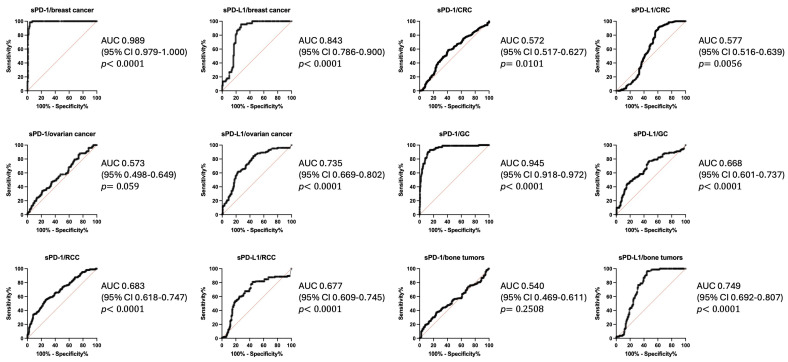
Receiver Operating Characteristic (ROC) curves for the diagnostic performance of sPD-1 and sPD-L1 levels in various malignancies. ROC analysis demonstrated the ability of serum sPD-1 and sPD-L1 levels to differentiate malignant from non-malignant cases across tumor types. sPD-1 shows excellent diagnostic accuracy in breast and gastric cancer, while sPD-L1 demonstrates moderate performance in breast and ovarian cancer. In other malignancies, both biomarkers exhibit low or no diagnostic value. The AUC values reflect the variable discriminatory power of these immune checkpoint molecules across different cancers. Red line indicates line of no discrimination, black line—receiver operating characteristic curve.

**Figure 3 biomedicines-13-00500-f003:**
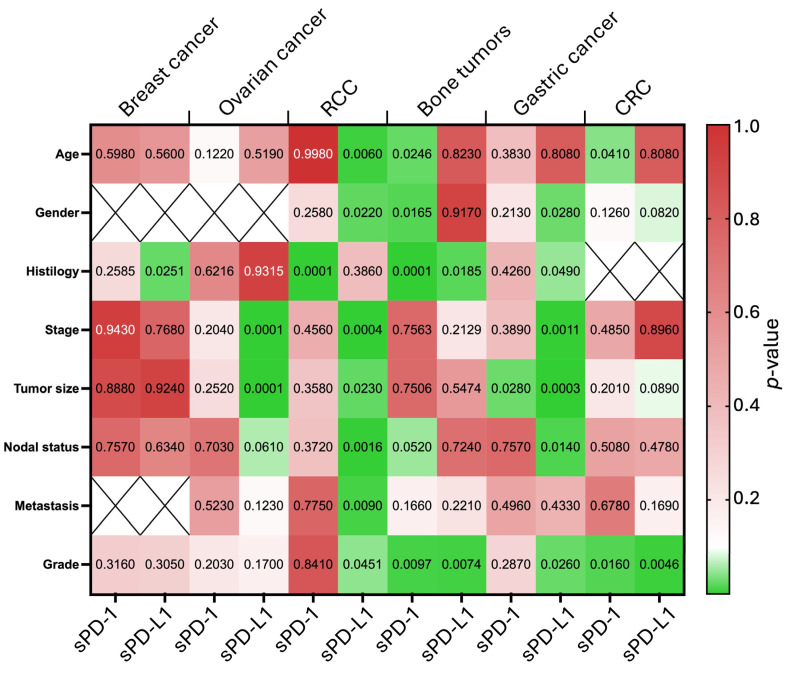
Heatmap summarizing statistical significance of associations between serum sPD-1 and sPD-L1 levels with clinicopathological characteristics across different tumor types. The heatmap represents the statistical significance of associations between serum sPD-1 and sPD-L1 levels and various tumor characteristics, including histological subtypes, stage, tumor size, nodal status, metastasis, and grade. Different tumor types are compared, highlighting variations in biomarker expression across malignancies. The color intensity reflects the strength of statistical significance, providing an overview of the tumor-specific patterns of sPD-1 and sPD-L1 regulation.

**Figure 4 biomedicines-13-00500-f004:**
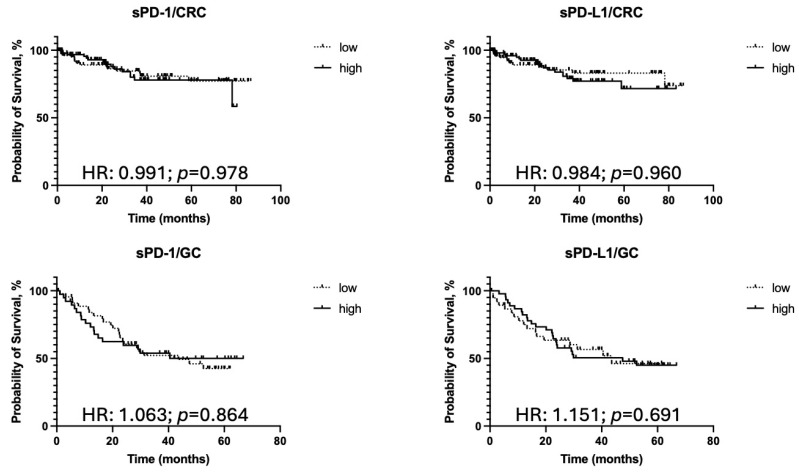
Overall survival (OS) analysis based on serum levels of sPD-1 and sPD-L1 across various malignancies. Kaplan–Meier survival curves show that high serum sPD-L1 levels are associated with significantly worse OS in renal cell carcinoma, ovarian cancer, and bone tumors. In contrast, sPD-1 levels generally do not impact OS, except in bone tumors, where lower levels are linked to a slightly improved prognosis. No significant associations were observed for gastric or colorectal cancer. *—statistically significant.

**Figure 5 biomedicines-13-00500-f005:**
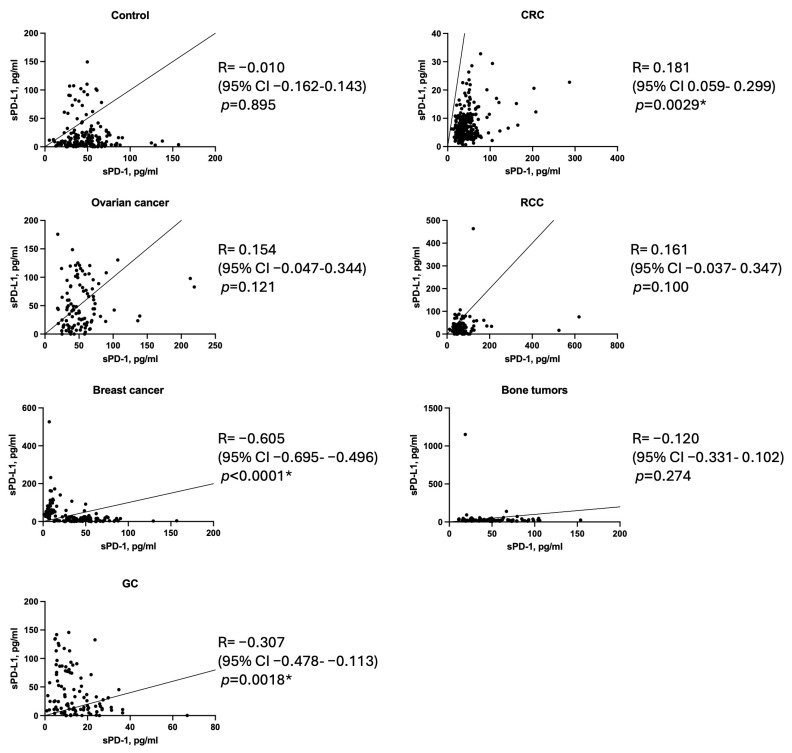
Correlation analysis of serum sPD-1 and sPD-L1 levels across various malignancies and healthy controls. A significant positive correlation was observed in colorectal cancer. In contrast, gastric cancer and breast cancer exhibited strong negative correlations. No significant correlation was found in ovarian cancer, RCC, bone tumors and healthy donors. *—statistically significant.

**Table 1 biomedicines-13-00500-t001:** Serum level of soluble sPD-1 and sPD-L1 in diverse tumors.

	sPD-1, pg/mL Median (IQR)	sPD-L1, pg/mL Median (IQR)
Colorectal cancer	38.92 (29.96–51.99)	6.78 (4.39–10.07)
Gastric cancer	11.82 (7.29–19.84)	24.55 (9.71–73.1)
Renal cancer	58.78 (44.05–76.99)	28.02 (12.18–43.79)
Bone tumors	47.88 (34.81–66.60)	22.86 (17.55–33.01)
Breast cancer	7.28 (5.84–9.04)	49.51 (39.1–81.86)
Ovarian cancer	48.20 (35.76–62.78)	42.37 (17.21–83.09)

**Table 2 biomedicines-13-00500-t002:** Association of serum sPD-1 and sPD-L1 levels with clinicopathological characteristics in breast cancer.

Characteristic	sPD-1, pg/mL			sPD-L1, pg/mL		
	Median	IQR	*p*	Median	IQR	*p*
Age						
≤49	7.46	6.15–9.09	0.598	50.58	35.02–82.35	0.560
>49	7.19	5.50–9.09		48.33	39.42–60.17	
Histology						
Luminal ^1^	6.81	5.28–9.39	^1^ vs. ^2^ > 0.9999	48.24	36.82–56.41	^1^ vs. ^2^ = 0.035 *
HER2+ ^2^	8.42	6.28–8.86	^1^ vs. ^3^ = 0.349	96.18	65.60–313.10	^1^ vs. ^3^ = 0.419
TNBC ^3^	7.89	6.47–9.53	^2^ vs. ^3^ > 0.9999	51.53	42.66–86.32	^2^ vs. ^3^ = 0.311
Stage						
II	7.47	6.26–8.68	0.943	50.25	38.31–79.81	0.768
III	7.17	5.80–9.54		48.52	39.10–81.86	
Tumor size						
T1–T2	7.46	5.26–8.96	0.888	49.33	39.06–77.37	0.924
T3–T4	7.23	5.84–9.17		50.25	38.78–81.86	
Nodal status						
N0	6.82	5.50–10.20	0.757	47.88	35.28–65.43	0.634
N+	7.46	5.84–9.04		50.63	39.10–82.05	
Grade						
G2	7.18	5.40–9.17	0.316	49.42	37.74–70.70	0.305
G3	7.66	6.49–8.95		56.76	41.48–105.90	

Superscript numbers indicate groups of comparison, *—statistically significant.

**Table 3 biomedicines-13-00500-t003:** Association of serum sPD-1 and sPD-L1 levels with clinicopathological characteristics in ovarian cancer.

Characteristic	sPD-1, pg/mL			sPD-L1, pg/mL		
	Median	IQR	*p*	Median	IQR	*p*
Age						
≤55	44.93	35.75–58.00	0.122	43.33	24.69–81.99	0.519
>55	51.85	37.81–69.50		40.62	13.45–84.70	
Histology						
Serous ^1^	48.09	37.38–64.48	^1^ vs. ^2^ > 0.9999	42.25	15.74–87.58	^1^ vs. ^2^ > 0.9999
Mucinous ^2^	40.55	32.60–58.81	^1^ vs. ^3^ = 0.349	44.88	18.95–75.78	^1^ vs. ^3^ = 0.349
Endometrioid ^3^	50.14	34.88–70.12	^2^ vs. ^3^ > 0.9999	42.01	20.74–62.53	^2^ vs. ^3^ > 0.9999
Stage						
I–II	47.87	35.31–58.69	0.204	17.88	6.87–44.88	<0.0001 *
III–IV	48.31	37.38–70.04		49.77	29.10–100.60	
Tumor size						
T1–T2	47.87	35.08–59.05	0.252	20.41	7.03–46.87	<0.0001 *
T3–T4	48.31	37.61–69.50		49.77	27.65–101.80	
Nodal status						
N0	48.09	35.65–63.27	0.703	42.01	15.74–80.03	0.061
N+	55.01	38.64–62.38		79.55	31.13–115.40	
Metastasis						
M0	47.87	35.76–61.85	0.523	42.01	15.93–77.09	0.123
M+	55.23	31.05–78.81		81.99	23.60–99.35	
Grade						
G1–G2	47.87	34.16–59.68	0.203	40.54	15.15–19.49	0.170
G3	50.33	38.53–67.91		44.41	69.00–100.60	

Superscript numbers indicate groups of comparison, *—statistically significant.

**Table 4 biomedicines-13-00500-t004:** Association of serum sPD-1 and sPD-L1 levels with clinicopathological characteristics in renal cell carcinoma.

Characteristic	sPD-1, pg/mL			sPD-L1, pg/mL		
	Median	IQR	*p*	Median	IQR	*p*
Age						
≤60	58.24	44.34–80.50	0.998	34.04	13.81–48.09	0.006 *
>60	61.96	41.80–74.84		18.08	8.45–34.50	
Gender						
male	60.63	45.84–79.86	0.258	32.55	12.18–46.92	0.022 *
female	53.88	38.62–74.80		17.88	8.45–37.08	
Histology						
ccRCC	55.37	40.62–72.64	<0.0001 *	26.84	12.10–41.16	0.386
nccRCC	87.01	74.39–123.2		33.65	12.18–50.04	
Stage						
I–II	56.66	41.53–79.86	0.456	20.91	12.18–33.45	0.0004 *
III–IV	62.51	48.02–76.36		38.37	23.82–67.22	
Tumor size						
T1–T2	56.15	42.4–76.97	0.358	24.26	12.18–33.26	0.023 *
T3–T4	62.82	50.23–77.97		37.55	17.7–66.53	
Nodal status						
N0	57.59	43.5–75.87	0.372	24.26	11.58–37.75	0.0016 *
N+	64.65	48.47–86.66		42.62	32.09–77.17	
Metastasis						
M0	58.42	41.82–80.34	0.775	24.87	11.01–38.37	0.009 *
M+	62.82	51.18–67.16		40.67	30.34–54.61	
Grade						
G1–G2	56.94	42.69–77.97	0.841	24.70	10.72–35.99	0.0451 *
G3–G4	61.29	39.77–72.71		32.32	18.41–54.68	

*—statistically significant.

**Table 5 biomedicines-13-00500-t005:** Association of serum sPD-1 and sPD-L1 levels with clinicopathological characteristics in bone tumors.

Characteristic	sPD-1, pg/mL			sPD-L1, pg/mL		
	Median	IQR	*p*	Median	IQR	*p*
Age						
≤40	57.11	37.40–69.89	0.0246 *	23.41	16.50–29.09	0.823
>40	40.70	25.19–57.94		18.08	17.57–38.12	
Gender						
male	52.02	37.34–67.78	0.0165 *	24.24	15.28–33.07	0.917
female	39.93	22.32–62.42		21.69	18.11–33.20	
Histology			^1^ vs. ^2^ < 0.0001 *			^1^ vs. ^2^ = 0.044 *
Osteosarcoma ^1^	61.39	46.43–76.84	^1^ vs. ^3^ > 0.9999	18.96	14.16–26.40	^1^ vs. ^3^ > 0.9999
Chondrosarcoma ^2^	36.25	24.90–54.62	^1^ vs. ^4^ = 0.359	26.63	19.72–40.19	^1^ vs. ^4^ > 0.9999
Ewing’s sarcoma ^3^	60.86	47.88–68.95	^2^ vs. ^3^ = 0.086	25.77	18.88–45.32	^2^ vs. ^3^ > 0.9999
Chordoma ^4^	49.9	38.96–55.60	^2^ vs. ^4^ > 0.9999	18.67	12.74–24.53	^2^ vs. ^4^ = 0.134
			^3^ vs. ^4^ > 0.9999			^3^ vs. ^4^ > 0.9999
Stage						
I ^1^	38.06	25.94–65.35	^1^ vs. ^2^ > 0.9999	35.62	19.62–47.09	^1^ vs. ^2^ = 0.237
II ^2^	47.88	35.53–64.15	^1^ vs. ^3^ > 0.9999	21.31	16.11–31.61	^1^ vs. ^3^ = 0.736
III–IV ^3^	55.47	23.99–80.42	^2^ vs. ^3^ > 0.9999	23.58	18.87–28.11	^2^ vs. ^3^ > 0.9999
Tumor size						
T1 ^1^	50.38	27.05–64.15	^1^ vs. ^2^ > 0.9999	21.11	18.13–39.03	^1^ vs. ^2^ > 0.9999
T2 ^2^	47.88	36.25–66.22	^1^ vs. ^3^ > 0.9999	22.58	15.01–32.91	^1^ vs. ^3^ > 0.9999
T3 ^3^	36.95	19.56–81.84	^2^ vs. ^3^ > 0.9999	26.24	20.19–33.82	^2^ vs. ^3^ = 0.909
Nodal status						
N0	47.50	33.86–65.16	0.052	22.86	17.55–33.81	0.724
N+	84.98	56.80–98.73		29.88	18.67–41.09	
Metastasis						
M0	47.50	30.85–65.16	0.166	23.96	17.55–36.34	0.221
M+	56.80	45.79–91.81		20.49	13.16–24.00	
Grade						
G1–G2	38.76	24.97–60.95	0.0097 *	26.24	20.00–40.69	0.0074 *
G3	54.15	38.75–68.80		20.32	15.10–28.36	

Superscript numbers indicate groups of comparison, *—statistically significant.

**Table 6 biomedicines-13-00500-t006:** Association of serum sPD-1 and sPD-L1 levels with clinicopathological characteristics in gastric cancer.

Characteristic	sPD-1, pg/mL			sPD-L1, pg/mL		
	Median	IQR	*p*	Median	IQR	*p*
Age						
≤60	11.82	7.74–20.85	0.383	26.08	10.06–71.93	0.808
>60	11.84	5.71–17.17		16.62	9.11–75.10	
Gender						
male	10.63	6.31–18.17	0.213	31.76	10.85–81.81	0.028 *
female	13.07	7.74–20.51		13.55	6.47–50.96	
Histology						
Adenocarcinoma	11.57	7.71–20.55	0.426	17.23	7.31–72.27	0.049 *
Signet ring cell carcinoma	11.97	5.51–17.15		46.45	16.20–75.60	
Stage						
I–II	12.11	9.39–24.09	0.389	10.98	6.47–27.61	0.0011 *
III–IV	11.45	6.86–18.28		36.47	13.27–81.81	
Tumor size						
T1–T2	15.62	10.18–26.00	0.028 *	10.06	4.66–15.96	0.0003 *
T3–T4	10.81	6.44–18.13		32.87	10.92–78.28	
Nodal status						
N0	11.60	6.04–19.89	0.757	14.39	6.73–50.07	0.014 *
N+	12.12	7.68–20.54		32.87	11.73–85.85	
Metastasis						
M0	11.82	6.77–21.07	0.496	20.65	8.79–73.10	0.433
M+	11.45	8.26–16.94		32.87	14.50–71.06	
Grade						
G1–G2	12.57	8.39–23.48	0.287	10.09	4.32–74.26	0.026 *
G3	11.29	6.86–18.60		29.31	11.37–72.85	

*—statistically significant.

**Table 7 biomedicines-13-00500-t007:** Association of serum sPD-1 and sPD-L1 levels with clinicopathological characteristics in colorectal cancer.

Characteristic	sPD-1, pg/mL			sPD-L1, pg/mL		
	Median	IQR	*p*	Median	IQR	*p*
Age						
≤62	35.56	28.64–49.95	0.041 *	6.15	3.93–9.05	0.808
>62	40.53	30.76–55.25		7.48	4.80–10.44	
Gender						
male	41.05	30.56–53.24	0.126	7.09	4.11–10.45	0.082
female	35.72	28.87–51.89		6.64	4.13–9.20	
Localization						
Left	40.22	29.61–53.09	0.977	6.78	4.39–10.07	0.644
Right	36.28	30.42–48.93		6.42	4.14–9.51	
Stage						
I–II	37.10	28.81–53.02	0.485	6.78	4.39–10.07	0.896
III–IV	40.43	30.71–51.94		6.68	4.09–9.65	
Tumor size						
T1–T2	44.47	31.01–57.049	0.201	5.95	4.26–8.95	0.089
T3–T4	37.11	29.79–51.18		7.24	4.41–10.12	
Nodal status						
N0	37.10	28.81–53.02	0.508	6.78	4.44–10.11	0.478
N+	40.74	30.95–51.94		6.52	3.88–9.48	
Metastasis						
M0	39.21	29.82–51.95	0.678	7.00	4.23–10.07	0.169
M+	37.09	30.20–50.86		5.55	3.79–9.49	
Grade						
G1	45.90	31.01–59.99	0.016 *	5.08	3.44–9.19	0.0046 *
G2–G3	36.30	29.51–49.94		7.34	4.79–10.19	

*—statistically significant.

**Table 8 biomedicines-13-00500-t008:** Univariate and multivariate analyses of overall survival.

Metrics		Univariate Analysis			Multivariate Analysis		
		HR	95% CI	*p*	HR	95% CI	*p*
sPD-1 (high/low)	ovarian	0.998	(0.463–2.153)	0.996	0.990	(0.947–1.002)	0.162
sPD-L1 (high/low)		2.366	(1.179–4.750)	0.017 *	1.009	(1.000–1.018)	0.038 *
sPD-1 (high/low)	renal	1.241	(0.526–2.921)	0.623	0.991	(0.965–1.010)	0.417
sPD-L1 (high/low)		4.343	(1.846–10.22)	0.004 *	1.005	(0.988–1.016)	0.368
sPD-1 (high/low)	bone	1.520	(0.791–2.923)	0.211	1.009	(0.994–1.022)	0.224
sPD-L1 (high/low)		1.304	(0.647–2.776)	0.428	1.053	(1.000–1.004)	0.010 *
sPD-1 (high/low)	gastric	1.063	(0.525–2.152)	0.864	0.987	(0.945–1.020)	0.484
sPD-L1 (high/low)		1.151	(0.568–2.335)	0.691	0.995	(0.985–1.003)	0.227
sPD-1 (high/low)	CRC	0.991	(0.535–1.836)	0.978	0.995	(0.974–1.008)	0.564
sPD-L1 (high/low)		0.984	(0.526–1.840)	0.960	1.037	(0.958–1.113)	0.337

*—statistically significant.

## Data Availability

Original data are available on reasonable request.

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
