# Peer review of "Assessing the Clinical Relevance of Soluble PD-1 and PD-L1: A Multi-Cohort Study Across Diverse Tumor Types and Prognostic Implications"

_biomedicines, 2025, doi:10.3390/biomedicines13020500_

Round 1
Reviewer 1 Report
Comments and Suggestions for Authors
This manuscript presents a comprehensive study on the clinical relevance of soluble PD-1 (sPD-1) and soluble PD-L1 (sPD-L1) in cancer patients, covering multiple malignancies and evaluating their diagnostic and prognostic value.
INTRODUCTION
The introduction provides a solid background.
METHODS
The study includes diverse malignancies, but the criteria for selecting specific cancers should be explained in more detail.
DISCUSSION
· sPD-L1 is a robust prognostic marker for ovarian and renal cancers but remains weak in other malignancies. The authors should explicitly discuss inconsistencies across different tumor types.
- The clinical applicability of sPD-L1 as a biomarker is emphasized, but challenges such as cost, reproducibility, and integration into clinical practice should be addressed.
- Possible biological mechanisms leading to variable sPD-1 and sPD-L1 levels across tumor types should be further explored.
· Across the manuscript, terms like “diagnostic utility” and “prognostic significance” should be clearly distinguished.
ENGLISH
Minor language errors should be corrected.
Author Response
We would like to thank the reviewer for valuable comments. We revised the manuscript accordingly as follows.
The study includes diverse malignancies, but the criteria for selecting specific cancers should be explained in more detail. – Additional description is added in the section 3.1.
The clinical applicability of sPD-L1 as a biomarker is emphasized, but challenges such as cost, reproducibility, and integration into clinical practice should be addressed. – A new paragraph is added in the discussion section.
Possible biological mechanisms leading to variable sPD-1 and sPD-L1 levels across tumor types should be further explored. – A new paragraph is added in the discussion section.
Across the manuscript, terms like “diagnostic utility” and “prognostic significance” should be clearly distinguished. - Corrected.
Minor language errors should be corrected. – The manuscript was spellchecked.
Reviewer 2 Report
Comments and Suggestions for Authors
In the current comprehensive study, Academician Kushlinskii N. E. with his colleagues, including Dr. Kovaleva O. V. and research group of prof. Gratchev A.N., conducted a large-scale evaluation of the levels of soluble forms of key immune checkpoints PD-1 and PD-L1 in the serum of patients with various oncological diseases, including gynecological cancers (breast and ovarian cancer), immunogenic kidney cancer, and rare tumors such as osteosarcoma and chondrosarcoma. The researchers evaluated the association of sPD-1 and sPD-L1 levels with a number of characteristics related to the severity of the disease course and obtained a number of important and new results that allow considering these proteins for monitoring and prognosis of the mentioned cancers (for example, for the first time it was found that high sPD-L1 level showed significant correlation with progression and poor prognosis of ovarian cancer). In my opinion, the obtained results are of high value for oncological diagnostics and undoubtedly deserve to be published in Biomedicines. The authors should only pay attention to the following minor comments:
(1) Dear authors, could you please summarize all the information obtained on different tumor types and characteristics in a final heat map (by p-value)? Such a presentation of the data would allow readers to have a more comprehensive look at all the data you have obtained.
(2) Line 55 - please convert PMID: 17304234 to a link
(3) Line 227 - please change the period after "Similarly" in the comma.
(4) line 357 - please put p in italics in p-values
Author Response
We would like to thank the reviewer for valuable comments. We revised the manuscript accordingly as follows.
(1) Dear authors, could you please summarize all the information obtained on different tumor types and characteristics in a final heat map (by p-value)? Such a presentation of the data would allow readers to have a more comprehensive look at all the data you have obtained. – Additional figure (Figure 3) added.
(2) Line 55 - please convert PMID: 17304234 to a link - corrected
(3) Line 227 - please change the period after "Similarly" in the comma. - corrected
(4) line 357 - please put p in italics in p-values - corrected
Reviewer 3 Report
Comments and Suggestions for Authors
· Information on antibody specificity, sensitivity, and validation against established standards for the ELISA experiment is missing
· The method for determining “high” vs. “low” sPD-1 and sPD-L1 levels is not clearly defined.
· There is repetitive introduction of the abbreviations for sPD-1/sPD-L1. Only mention once at the beginning of the paper and use the abbreviated versions only after that.
· "Interferon-gamma (IFN-γ), in particular, has been shown to induce PD-L1 expression in ovarian cancer cells, facilitating disease progression. Notably, inhibition of the IFN-γ receptor has been demonstrated to reduce PD-L1 expression, suggesting a therapeutic avenue for modulating this immune checkpoint pathway. When PD-L1 binds to PD-1, it triggers PD-1-mediated signaling pathways within T cells, leading to the suppression of their proliferation, activation, and survival. This signaling cascade effectively dampens the immune response, allowing tumor cells to evade immune-mediated destruction. The downstream effects include reduced cytokine production, metabolic reprogramming of T cells, and the promotion of an exhausted T-cell phenotype, which is characterized by diminished effector function [6]." This paragraph is highlighting multiple points and using only one reference. Address the reference for each sentence clearly rather than gathering them all in only one reference.
· Add explanation on how were confounding factors accounted for during patient selection.
· Explain why Mann–Whitney U test and Kruskal–Wallis test were used instead of parametric tests.
· The "healthy controls" are described as age- and sex-matched but not screened for subclinical inflammatory conditions, which could confound comparisons with cancer patients.
· The transition from the membrane-bound version of PD-1/PD-L1 to soluble forms is too abrupt, failing to provide rationale for why soluble versions might be better for biomarkers or provide complementary insight.
· Figure captions and legends are not detailed enough to allow standalone interpretation. Include information on sample size, statistical methods, and different panels.
· The discussion could be strengthened by explicitly addressing the clinical utility of these findings. How might the measurement of sPD-1/sPD-L1 levels be integrated into existing clinical practice? Could it improve patient stratification, treatment selection, or prognosis prediction? The authors could provide concrete suggestions on how these findings translate into actionable clinical strategies.
· Though the discussion mentions sPD-L1 as a marker, it does not even touch on the aspect of whether it could predict responses to immune checkpoint inhibitors.
· The paper lacks a comparative analysis of sPD-1/sPD-L1 with established biomarkers for the same cancer types. How do the diagnostic and prognostic values of sPD-1/sPD-L1 compared to currently used markers? This comparison is vital for establishing the added value and clinical relevance of the proposed biomarkers.
· The use of serum levels as a proxy for tumor microenvironment dynamics is not evaluated. Also, expand the limitations section to include the heterogeneity of patient cohorts.
Author Response
We would like to thank the reviewer for valuable comments. We revised the manuscript accordingly as follows.
Information on antibody specificity, sensitivity, and validation against established standards for the ELISA experiment is missing -Additional information is provided in the methods section 2.2.
- The method for determining “high” vs. “low” sPD-1 and sPD-L1 levels is not clearly defined. – Additional info is provided in the section 2.3.
- There is repetitive introduction of the abbreviations for sPD-1/sPD-L1. Only mention once at the beginning of the paper and use the abbreviated versions only after that. – Corrected.
- "Interferon-gamma (IFN-γ), in particular, has been shown to induce PD-L1 expression in ovarian cancer cells, facilitating disease progression. Notably, inhibition of the IFN-γ receptor has been demonstrated to reduce PD-L1 expression, suggesting a therapeutic avenue for modulating this immune checkpoint pathway. When PD-L1 binds to PD-1, it triggers PD-1-mediated signaling pathways within T cells, leading to the suppression of their proliferation, activation, and survival. This signaling cascade effectively dampens the immune response, allowing tumor cells to evade immune-mediated destruction. The downstream effects include reduced cytokine production, metabolic reprogramming of T cells, and the promotion of an exhausted T-cell phenotype, which is characterized by diminished effector function [6]." This paragraph is highlighting multiple points and using only one reference. Address the reference for each sentence clearly rather than gathering them all in only one reference. – Citations added.
- Add explanation on how were confounding factors accounted for during patient selection. – Inclusion and exclusion criteria were added to the section 2.1.
- Explain why Mann–Whitney U test and Kruskal–Wallis test were used instead of parametric tests. – Section 2.3 updated.
- The "healthy controls" are described as age- and sex-matched but not screened for subclinical inflammatory conditions, which could confound comparisons with cancer patients. – Additional information provided in the section 2.1.
- The transition from the membrane-bound version of PD-1/PD-L1 to soluble forms is too abrupt, failing to provide rationale for why soluble versions might be better for biomarkers or provide complementary insight. - We did not claim that soluble forms of PD-1/PD-L1 might be better biomarkers or provide complementary insight. There is also a transition from membrane bound to soluble forms in the lines 75-76.
- Figure captions and legends are not detailed enough to allow standalone interpretation. Include information on sample size, statistical methods, and different panels. – Captions and legends extended.
- The discussion could be strengthened by explicitly addressing the clinical utility of these findings. How might the measurement of sPD-1/sPD-L1 levels be integrated into existing clinical practice? Could it improve patient stratification, treatment selection, or prognosis prediction? The authors could provide concrete suggestions on how these findings translate into actionable clinical strategies. – We added a paragraph on the challenges of integration of this testing in clinical practice.
- Though the discussion mentions sPD-L1 as a marker, it does not even touch on the aspect of whether it could predict responses to immune checkpoint inhibitors. – We added short paragraph about the importance of sPD-L1 and sPD-1 in the discussion.
- The paper lacks a comparative analysis of sPD-1/sPD-L1 with established biomarkers for the same cancer types. How do the diagnostic and prognostic values of sPD-1/sPD-L1 compared to currently used markers? This comparison is vital for establishing the added value and clinical relevance of the proposed biomarkers. – Additional paragraph added in the discussion section.
- The use of serum levels as a proxy for tumor microenvironment dynamics is not evaluated. Also, expand the limitations section to include the heterogeneity of patient cohorts. – Section on study limitations is added.
Round 2
Reviewer 3 Report
Comments and Suggestions for Authors
Thank you for considering all comments, the manuscript is better now.